# HEAT: Hyperedge Attention Networks

**Dobrik Georgiev**[*]                    *dgg30@cam.ac.uk*
*Department of Computer Science and Technology, University of Cambridge, UK*

**Marc Brockschmidt**                    *mmjb@google.com*
*Microsoft Research, Cambridge, UK*[†]

**Miltiadis Allamanis**                    *mallamanis@google.com*
*Microsoft Research, Cambridge, UK*[†]

**Reviewed on OpenReview:** *https://openreview.net/forum?id=gCmQK6McbR*

## Abstract

Learning from structured data is a core machine learning task. Commonly, such data is represented as graphs, which normally only consider (typed) binary relationships between pairs of nodes. This is a substantial limitation for many domains with highly-structured data. One important such domain is source code, where hypergraph-based representations can better capture the semantically rich and structured nature of code.

In this work, we present HEAT, a neural model capable of representing typed and qualified hypergraphs, where each hyperedge explicitly qualifies how participating nodes contribute. It can be viewed as a generalization of both message passing neural networks and Transformers. We evaluate HEAT on knowledge base completion and on bug detection and repair using a novel hypergraph representation of programs. In both settings, it outperforms strong baselines, indicating its power and generality.

## 1 Introduction

Large parts of human knowledge can be formally represented as sets of relations between entities, allowing for mechanical reasoning over it. Common examples of this view are knowledge graphs representing our environment, databases representing business details, and first-order formulas describing mathematical insights. Such structured data hence regularly appears as input to machine learning (ML) systems.

In practice, this very generic framework is not easy to handle in ML models. One issue is that the set of relations is not necessarily known beforehand, and that the precise structure of a relation is not easily fixed. As an example, consider `studied(person:`$P$`, institution:`$I$`, major:`$M$`)`, encoding the fact that a person $P$ studied at institution $I$, majoring in $M$. However, if the institution is unknown, we may want to just consider `studied(person:`$P$`, major:`$M$`)`, or we may need to handle case of people double-majoring, using `studied(person:`$P$`, major:`$M_2$`, major:`$M_1$`)`.

Existing ML approaches usually cast such data as hypergraphs, and extend the field of graph learning approaches to this setting, but struggle with its generality. Some solutions require to know the set of relations beforehand (fixing their arity, and assigning fixed meanings to each parameter), while others abstract the set of entities to a simple set and forego the use of the name of the relation and its parameters.

---

[*]Work mainly performed while interning at Microsoft Research, Cambridge, UK.
[†]Now at Google Research.

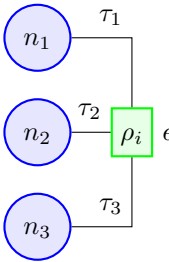

Figure 1: Representation of relation $\rho(\tau_1\!:\!n_1,\ \tau_2\!:\!n_2,\ \tau_3\!:\!n_3)$ as typed and qualified hyperedge $e$, *i.e.* a relation of type $\rho$, and the qualified participation of the nodes $n_1$ as $\tau_1$, $n_2$ as $\tau_2$, and $n_3$ as $\tau_3$ in $e$. HEAT operates on hypergraphs with such hyperedges.

One form of data that can profit from being modeled as a hypergraph is program source code. Existing approaches model code either as simple sequence of tokens (Hindle et al., 2012) or as a graph (Allamanis et al., 2018b; Hellendoorn et al., 2020). While the former can easily leverage successful techniques from the NLP domain, it is not possible to include additional domain-specific knowledge (such as the flow of data) in the input. On the other hand, graph models are able to take some of this additional information, but struggle to represent more complex relationships that require hyperedges.

In this work, we propose a new architecture, HEAT (HyperEdge ATtention), that is able to handle an open set of relations that may appear in several variants. To this end, we combine the idea of message passing schemes that follow the graph structure with the flexibility and representational power of Transformer architectures. Concretely, our model handles arbitrary relations by presenting each one as a separate sequence in a Transformer, where an idea akin to standard positional encodings is used to represent how each entity participates in the relation. The Transformer output is then used to update the representations of participating entities.

We illustrate the success of this technique in two very different settings: learning to infer additional relations in knowledge graphs, and learning to find and repair bugs in programs. In both cases, our model shows improvements over strong state-of-the-art methods. Concretely, we (a) define HEAT as a novel hypergraph neural network architecture (Sec. 2), (b) define a novel representation of programs as hypergraphs (Sec. 3), (c) evaluate HEAT on the tasks of detecting and repairing bugs (Sec. 4.1) and link prediction in knowledge graphs (Sec. 4.2).

Our implementation of the HEAT model is available on the `heat` branch of `https://github.com/microsoft/neurips21-self-supervised-bug-detection-and-repair/tree/heat`. This includes code for the extraction of hypergraph representations of Python code as discussed in Sec. 3.

## 2   The HEAT Model

In this work, we are interested in representing typed and qualified relations of the form `relName(parName1:`$n_1$`, parName2:`$n_2$`, ...)` where `relName` represents the name of a relation, $n_1, n_2, \ldots$ are entities participating in the relation, with `parName1`, ... describes (qualifies) their role in the relation.

Formally, such relations can be represented as *typed* and *qualified* hypergraphs: we consider a set of nodes (entities) $\mathcal{N} = \{n_1, \ldots\}$ and a set of hyperedges $\mathcal{H} = \{e_1, \ldots\}$. Each hyperedge $e = (\rho, \{(\tau_1, n_1), \ldots, (\tau_k, n_k)\})$ describes a named relationship of type $\rho$ among nodes $n_1 \ldots n_k$ where the role of node $n_k$ within $e$ is qualified using $\tau_k$. Fig. 1 illustrates the form of such a hyperedge. Note that this is in contrast to traditional hypergraphs where nodes participate in a hyperedge without any qualifiers. Instead, typed and qualified graphs can accurately represent a large range of domains maintaining valuable information.

### 2.1 Background

Message passing neural networks (MPNN) (Gilmer et al., 2017) operate on sets of nodes $\mathcal{N}$ and sets of (typed) edges $\mathcal{E}$, where each edge $(n_i, \tau_i, n_j)$ of type $\tau$ connects a pair of nodes. In MPNNs, each node $n \in \mathcal{N}$ is associated with representations $\mathbf{h}_n^{(t)}$ that are computed incrementally. The initial $\mathbf{h}_n^{(0)}$ stems from input features, but subsequent representations are computed by exchanging information between nodes. Optionally, each edge $e$ may also be associated with a state/representation $\mathbf{h}_e^{(t)}$ that is also computed incrementally. Concretely, each edge gives rise to a "message" using a learnable function $f_m$:

$$\mathbf{m}_{(n_i, \tau, n_j)}^{(t)} = f_m \left( \mathbf{h}_{n_i}^{(t)}, \tau, \mathbf{h}_{n_j}^{(t)}, \mathbf{h}_e^{(t)} \right).$$

To update the representation of a node $n$, all incoming messages are aggregated with a permutation-invariant function $\text{AGG}(\cdot)$ into a single update, *i.e.*

$$\mathbf{u}_{n_j}^{(t)} = \text{AGG} \left( \left\{ \mathbf{m}_{(n_i, \tau, n_j)}^{(t)} \mid (n_i, \tau, n_j) \in \mathcal{E} \right\} \right). \tag{1}$$

$\text{AGG}$ is commonly implemented as summation or max-pooling, though attention-based variants exist (Veličković et al., 2018). Finally, each node's representation is updated using its original representation and the aggregated messages. Many different update mechanisms have been considered, ranging from just using the aggregated messages (Kipf & Welling, 2017) to gated update functions (Li et al., 2016).

Transformers (Vaswani et al., 2017) learn representations of sets of elements $\mathcal{N}$ without explicit edges. Instead, they consider all pairwise interactions and use a learned attention mechanism to identify particularly important pairs. Transformer layers are split into two sublayers. First, multihead attention (MHA) uses the representations of all $\mathcal{N}$ to compute an "update" $\mathbf{u}_n$ for each $n \in \mathcal{N}$, *i.e.*

$$\left[ \mathbf{u}_{n_1}^{(t)}, \ldots, \mathbf{u}_{n_k}^{(t)} \right] = \text{MHA} \left( \left[ \mathbf{h}_{n_1}^{(t)}, \ldots, \mathbf{h}_{n_k}^{(t)} \right] \right). \tag{2}$$

Internally, MHA uses an attention mechanism to determine the relative importance of pairs of entities and to compute the updated representations. Note that MHA treats its input as a (multi-)set, and hence is permutation-invariant. To provide ordering information, a common approach is to use positional encodings, *i.e.* to extend each representation with explicit information about the position. In practice, this means that MHA operates on $\left[ \mathbf{h}_{n_1}^{(t)} \oplus \mathbf{p}_1, \ldots, \mathbf{h}_{n_k}^{(t)} \oplus \mathbf{p}_k \right]$, where $\mathbf{p}_i$ provides information about the position of $n_i$, and $\oplus$ is a combination operation (commonly element-wise addition).

The second Transformer sublayer is used to combine the updated representations with residual information and add computational depth to the model. This is implemented as

$$\mathbf{q}_{n_i}^{(t)} = \text{LN} \left( \mathbf{h}_{n_i}^{(t)} + \mathbf{u}_{n_i}^{(t)} \right) \tag{3}$$

$$\mathbf{h}_{n_i}^{(t+1)} = \text{LN} \left( \mathbf{q}_{n_i}^{(t)} + \text{FFN} \left( \mathbf{q}_{n_i}^{(t)} \right) \right), \tag{4}$$

where LN is layer normalisation (Ba et al., 2016) and FFN is a feedforward neural network (commonly with one large intermediate hidden layer).

### 2.2 HEAT: An Attention-Based Hypergraph Neural Network

We now present HEAT, a Transformer-based message passing hypergraph neural network that learns to represent typed and qualified hypergraphs.

Overall, we follow the core idea of the standard message passing paradigm: our aim is to update the representation of each entity using messages arising from its relations to other entities. Following Battaglia et al. (2018), we also explicitly consider representations of each hyperedge, based on their type and the representation of adjacent entities. Intuitively, we want each entity to "receive" one message per hyperedge it participates in, reflecting both its qualifier (*i.e.* role) in that hyperedge, as well as the relation to other participating entities.

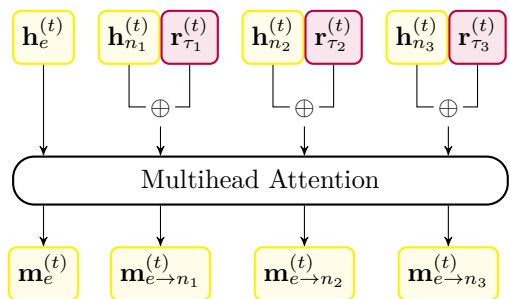

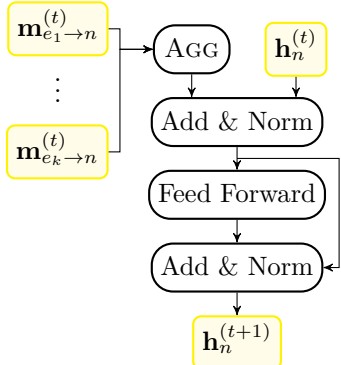

(a) Message computation for hyperedge $e$ from Fig. 1. The previous node states $\mathbf{h}_{n_i}^{(t)}$ are combined with their respective qualifier embeddings $\mathbf{r}_{\tau_i}^{(t)}$. Then, a multihead attention mechanism computes the messages $\mathbf{m}_{e\to\cdot}^{(t)}$ and $\mathbf{m}_e^{(t)}$.

(b) The state update for each node $n$ in the hypergraph. Apart from AGG this matches a standard Transformer layer of Vaswani et al. (2017).

Figure 2: The HEAT Architecture.

To compute these messages, we borrow ideas from the Transformer architecture. We view each qualified hyperedge as a set of entities, with each entity being associated with a position (its qualifier in the hyperedge), and then use multihead attention — as in Eq. 2 — to compute one message per involved entity:

$$\left[\mathbf{m}_e^{(t)}, \mathbf{m}_{e\to n_1}^{(t)}, \ldots, \mathbf{m}_{e\to n_k}^{(t)}\right] = \mathrm{MHA}\left(\left[\mathbf{h}_e^{(t)}, \mathbf{r}_{\tau_1}^{(t)} \oplus \mathbf{h}_{n_1}^{(t)}, \ldots, \mathbf{r}_{\tau_k}^{(t)} \oplus \mathbf{h}_{n_k}^{(t)}\right]\right). \tag{5}$$

Here, $\mathbf{h}_e^{(t)}$ is the current representation of the hyperedge, $\mathbf{h}_{n_i}^{(t)}$ is the current representation of node $n_i$, $\mathbf{r}_{\tau_i}^{(t)}$ is the representation (embedding) of the qualifier $\tau_i$ which denotes the role of $n_i$ in $e$, and $\oplus$ combines two vectors. Fig. 2a illustrates this. In this work, we consider element-wise addition for $\oplus$ but other operators, such as vector concatenation would be possible.

We can then compute a single update $\mathbf{u}_{n_i}^{(t)}$ for node $n_i$ by aggregating the relevant messages computed for the hyperedges it occurs in, *i.e.*

$$\mathbf{u}_{n_i}^{(t)} = \mathrm{AGG}\left(\left\{\mathbf{m}_{e\to n_i}^{(t)} \mid e = (\rho, \{\ldots, (\tau_i, n_i), \ldots\}) \in \mathcal{H}\right\}\right).$$

This is equivalent to the aggregation of messages in MPNNs (cf. Eq. 1). We then compute an updated entity representation $\mathbf{h}_{n_i}^{(t+1)}$ following the Transformer update as described in Eqs. 3, 4. This is illustrated in Fig. 2b. The core difference to the Transformer case is the aggregation $\mathrm{AGG}(\cdot)$ used to combine all messages (as in a message passing network). $\mathrm{AGG}(\cdot)$ can be any permutation invariant function, such as elementwise-sum and max, or even an another transformer (see Sec. 4.1 for experimental evaluation of these variants).

Finally, the hyperedge states $\mathbf{h}_e^{(t)}$ are also updated as in Eqs. 3, 4 using the messages $\mathbf{m}_e^{(t)}$ computed as in Eq. 5, and no aggregation is required because there is only a single message for each hyperedge $e$. Initial node states $\mathbf{h}_{n_i}^{(0)}$ are initialised with node-level information (as in GNNs). Qualifier embeddings $\mathbf{r}_{\tau_i}$ (resp. initial hyperedge states $\mathbf{h}_e^{(0)}$) are obtained by breaking the name of $\tau_i$ (resp. $\rho_e$) into subtokens (*e.g.*, `parName` is split into `par` and `name` or `foo_bar2` into `foo`, `bar`, and `2`), embedding these through a (learnable) vocabulary embedding matrix, and then sum-pooling. Then for each HEAT layer $\mathbf{r}_{\tau_i}^{(t)}$ is computed through a linear learnable layer of the sum-pooled embedding, *i.e.* $\mathbf{r}_{\tau_i}^{(t)} = W_\tau^{(t)}\mathbf{r}_{\tau_i} + \mathbf{b}_\tau^{(t)}$.

**Generalising Transformers** Note that if the hypergraph is made up of a single hyperedge of the form `Seq(pos1:`$n_1$`, pos2:`$n_2$`, ...)` then HEAT degenerates to the standard transformer of Vaswani et al. (2017) with positional encodings over the sequence $n_1, n_2, \ldots$ at each layer. In the case of such sequence positions, we simply use the sinusoidal positional encodings of Vaswani et al. (2017) as position embedding, rather than using a learnable embedding. Thus HEAT can be thought to generalise transformers from sets (and sequences) to richer structures.

**Computational Considerations** A naïve implementation of a HEAT message passing step requires one sample per hyperedge to be fed into a multihead attention layer, as each hyperedge of $k$ nodes gives rise to a sequence of length $k + 1$, holding the edge and all node representations. To enable efficient batching, this would require to pad all such sequences to the longest sequence present in a batch. However, the length of these sequences may vary wildly. On the other hand, processing each hyperedge separately would not make use of parallel computation in modern GPUs.

To resolve this, we use two tricks. First, we consider "microbatches" with a pre-defined set of sequence lengths $\{16, 64, 256, 768, 1024\}$, minimising wastage from padding. Second, we "pack" several shorter sequences into a single sequence, and then use appropriate attention masks in the MHA computation to avoid interactions. For example, a hyperedge of size 9 and a hyperedge of size 7 can be joined together to yield a sequence of length 16. Appx. A details the packing algorithm used. This process is performed in-CPU during the minibatch preparation.

## 3 Representing Code as Hypergraphs

Program source code is a highly structured object that can be augmented with rich relations obtained from program analyses. Traditionally, source code is represented in machine learning either as a sequence of tokens (Hindle et al., 2012), a tree (Yin & Neubig, 2017), or as a graph (Allamanis et al., 2018b; Hellendoorn et al., 2020). Graph-based representations (subsuming tree-based ones) commonly consider pairwise relationships among entities in the code, whereas token-level ones simply consume a token sequence. In this section, we present a novel hypergraph representation of code that retains the best of both representations.

Our program hypergraph construction is derived from the graph construction presented by Allamanis et al. (2021), but uses hyperedges to produce a more informative and compact representation. The set of nodes in the generated hypergraphs include tokens, expressions, abstract syntax tree (AST) nodes, and symbols. However, in contrast to prior work, we do not only consider pairwise relationships, but instead use a typed and qualified hypergraph (Fig. 1). We detail the used hyperedges and their advantages next. Fig. 3 in Appx. B illustrates some of the considered relations on a small synthetic snippet.

**Tokens** For the token sequence $t_1, t_2, ..., t_L$ of a snippet of source code, we create the relation `Tokens(p1:`$t_1$`, p2:`$t_2$`, ...)`. This is the entire information used by token-level Transformer models, considering all-to-all relations among tokens. Note that in standard graph-based code representations, the token sequence is usually represented using a chain of `NextToken` binary edges, meaning that long-distance relationships are hard to discover for models consuming such graphs. For very long token sequences, which may cause memory issues, we "chunk" the sequence into overlapping segments of length $L$, similar to windowed, sparse attention approaches. We use $L = 512$ in the experiments. Within HEAT and specifically for the `Tokens(·)` hyperedges, to reduce the used parameters and to allow arbitrarily long sequences, the qualifier embeddings $\mathbf{r}_{\text{p1}}, \mathbf{r}_{\text{p2}}, \mathbf{r}_{\text{p3}}, ...$ are computed using the fixed sinusoidal embeddings of Vaswani et al. (2017) instead of learnable embeddings.

**AST** We represent the program's abstract syntax tree using `AstNode` relations, with qualifiers corresponding to the names of children of each node. For example, an AST node of a binary operation is represented as `AstNode(node:`$n_{\text{BinOp}}$`, left:`$n_{\text{left}}$`, op:`$n_{\text{op}}$`, right:`$n_{\text{right}}$`)`. Similarly, a `AstNode(node:`$n_{\text{IfStmt}}$`, cond:`$n_c$`, then:`$n_t$`, else:`$n_e$`)` represents an `if` statement node $n_{\text{IfStmt}}$. In cases where the children have the same qualifier and are ordered, (*e.g.* the sequential statements within a block) we create numbered relations, *i.e.* `AstNode(node:`$n_{\text{Block}}$`, s1:`$n_1$`, s2:`$n_2$`, ...)`. For Python, we use as qualifier names those used in libCST. In contrast, most graph-based approaches use `Child` edges to connect a parent node to all of its children, and thus lose information about the specific role of child nodes. As a consequence, this means that left and right children of non-commutative operations can not easily be distinguished. Allamanis et al. (2021) attempts to rectify this using a `NextSibling` edge type, but still is not able to make use of the known role of each child.

**Control and Data Flow** To indicate that program execution can flow from any of the AST nodes $p_1, p_2, ...$ to one of the AST nodes $s_1, s_2, ...$, we use the relation `CtrlF(prev:`$n_{p_1}$`, prev:`$n_{p_2}$`, ..., succ:`$n_{s_1}$`, succ:`$n_{s_2}$`)`. For example, `if c: p1 else: p2; s1` would yield `CtrlF(prev:p1, prev:p2, succ:s1)`. Similarly, we use relations `MayRead` and `MayWrite` to represent dataflow, where the previous location at which a

symbol may have been read (written to) are connected to the succeeding locations. Note that these relations compactly represent data and control flow consolidating $N$-to-$M$ relations (*e.g.* $N$ states may lead to one of $M$ states) into a single relation, which would otherwise require $N \cdot M$ edges in a standard graph.

**Symbols** We use the relation `Symbol(sym:`$n_s$`, occ:`$n_1$`, occ:`$n_2$`, ..., may_last_use:`$n_{u_1}$`,` `may_last_use:`$n_{u_2}$`, ... )` to connect all nodes $n_1, n_2,...$ referring to a symbol (*e.g.* variable) to a fresh node $n_s$, introduced for each occurring symbol. We annotate within this relation the nodes $n_{u_1}, n_{u_2},...$ that are the potential last uses of the symbol within the code snippet.

**Functions** A challenge in representing source code is how to handle calls to (potentially user-defined) functions. As it is usually not feasible to include the source code of all called functions in the model input (for computational and memory reasons), appropriate abstractions need to be used. For a call (invocation) of a function `foo(arg1, ..., argN)` defined by `def foo(par1, ..., parN): ...`, we introduce the relation `foo(rval:`$n$`, par1:`$n_\mathtt{arg1}$`, ..., par3:`$n_\mathtt{argN}$`)` where $n$ is the invocation expression node and $n_\mathtt{arg1}, \ldots, n_\mathtt{argN}$ are the nodes representing each of the arguments. Hence, we generate one relation symbol for each defined function, and match nodes representing arguments to the formal parameter names as qualifiers. This representation allows to naturally handle the case of variable numbers of parameters and arbitrary functions.

Syntactic sugar and operators are converted in the same way, using the name of the corresponding built-in function. For example, in Python, `a in b` is converted into the relation `__contains__(self:`$n_\mathtt{b}$`, item:`$n_\mathtt{a}$`)` and `a -= b` is converted into `__isub__(self:`$n_\mathtt{a}$`, other:`$n_\mathtt{b}$`)` following the reference Python data model.

Finally, we use the relation `Returns(fn:`$n_\mathtt{f}$`, from:`$n_1$`, from:`$n_2$`, ...)` to connect all possible return points for a function with the AST node $n_\mathtt{f}$ for the function definition. Similarly, a `Yields(·)` is defined for generator functions.

## 4 Evaluation

We evaluate HEAT on two tasks from the literature: bug detection and repair (Allamanis et al., 2021) and knowledge base completion (Galkin et al., 2020). We implemented it as a PyTorch (Paszke et al., 2019) `Module`, available on the `heat` branch of `https://github.com/microsoft/neurips21-self-supervised-bug-detection-and-repair/tree/heat`.

### 4.1 HEAT for Bug Detection & Repair

We evaluate HEAT on the bug localisation and detection task of Allamanis et al. (2021) in the supervised setting. This is a hard task that requires combining ambiguous information with reasoning capabilities able to detect bugs in real-life source code.

For this, we built on the open-source release of PyBugLab of Allamanis et al. (2021), making two changes: (1) we adapt the graph construction from programs to produce hypergraphs as discussed in Sec. 3, and (2) use HEAT to compute entity representations from the generated graphs, rather than GNNs or GREAT.

**Dataset** We use the code of Allamanis et al. (2021) to generate a dataset of randomly inserted bugs to train and evaluate a neural network in a supervised fashion. Consequently, we obtain a new variant of the "Random Bugs" test dataset, consisting of $\sim 760k$ graphs. We additionally re-extract the PyPIBugs dataset with the provided script, generating hypergraphs as consumed by HEAT, and graphs generated by the baseline models. However, since the PyPIBugs dataset is provided in the form of GitHub URLs referring to the buggy commit SHAs, some of them have been removed from GitHub and thus our PyPIBugs dataset contains 2354 samples, 20 less than the one used by Allamanis et al. (2021).

**Model Architecture** We modify the architecture of Allamanis et al. (2021) to use 6 HEAT layers with hidden dimension of 256, 8 heads, feed-forward (FFN in Eq. 4) hidden layer of 2048, and dropout rate of 0.1. As discussed above, our datasets differ slightly from the data used by Allamanis et al. (2021), and we re-evaluated their released code on our new datasets. We found this rerun to perform notably better than what

Table 1: Evaluation Results on Supervised Bug Detection and Repair on supervised PyBugLab.

| | Random Bugs | | | PyPIBugs | | |
|---|---|---|---|---|---|---|
| | Joint | Loc. | Repair | Joint | Loc. | Repair |
| GNN (Allamanis et al., 2021)[†] | 62.4 | 73.6 | 81.2 | 20.0 | 28.4 | 61.8 |
| GREAT (Allamanis et al., 2021)[†] | 51.0 | 61.9 | 76.3 | 16.8 | 25.8 | 58.6 |
| GNN (Allamanis et al., 2021) (rerun) | 69.8 | 79.6 | 83.4 | 22.0 | 28.3 | 66.7 |
| GREAT (Allamanis et al., 2021) (rerun) | 65.6 | 74.4 | 81.8 | 16.8 | 21.9 | 67.7 |
| HEAT | **76.5** | **83.1** | **88.5** | **24.6** | **29.6** | 71.0 |
| HEAT – DeepSet-based messages | 74.0 | 81.1 | 87.3 | 23.2 | 28.6 | 69.0 |
| HEAT – without qualifier embeddings | 69.2 | 76.7 | 85.6 | 19.1 | 25.7 | 65.0 |
| HEAT – $\textsc{Agg} \triangleq$ CrossAtt | **76.5** | 83.0 | 88.2 | 23.4 | 27.9 | **71.5** |
| HEAT – without FFN | 73.7 | 80.7 | 87.0 | 20.7 | 25.4 | 69.2 |
| HEAT – without hyperedge state | 75.7 | 82.4 | 88.0 | 23.0 | 28.6 | 71.1 |

[†] Reported on a different random bugs dataset and a slightly different PyPIBugs dataset.

was originally reported in the paper. In private communication, the authors explained that their public code included a small change to the model architecture compared to the paper: the subnetworks used for selecting a program repair rule are now shallow MLPs (rather than inner products), which increases performance across the board. Our HEAT extension follows the code release, and hence we use max-pooled subtoken embeddings to initialise node embeddings, a pointer network-like submodel to select which part of the program to repair, and a shallow MLPs to select the repair rules. We also re-use the PyBugLab supervised objective, which is composed of two parts: a PointerNet-style objective requiring to identify the graph node corresponding to a program bug, and a ranking objective requiring to select a fixing program rewrite at the selected location.

**Results**  We show the results of our experiments in Table 1, where "Loc." refers to the accuracy in identifying the buggy location in an input program, "Repair" to the accuracy in determining the correct fix given the buggy location, and "Joint" to solving both tasks together. The results indicate that HEAT improves performance on both considered datasets, improving the joint localisation and repair accuracy by $\sim 10\%$ over the two well-tuned baselines.

In particular, we observe that HEAT substantially improves over GREAT (Hellendoorn et al., 2020), which also adapts the Transformer architecture to include relational information. However, GREAT eschews explicit modelling of edges, and instead uses (binary) relations between tokens to bias the attention weights in the MHA computation. We observe a less pronounced gain over GNNs, which we believe is due to the simpler information flow across long distances and the clearer way of encoding structural information in HEAT. (see Sec. 3) We note that Allamanis et al. (2021) showed that their models also outperform fine-tuned variants of the cuBERT (Kanade et al., 2020) model, which stems from self-supervised pre-training using masking objectives. Consequently, we believe that HEAT outperforms such large models as well, though a comparison to recent very large models, adapted to the task (Chen et al., 2021; Austin et al., 2021; Li et al., 2022) is left to future work.

**Variations and Ablations**  To understand the importance of different components of HEAT, we experiment with five ablations and variations, shown on the bottom of Table 1.

First, we study the importance of using multi-head attention to compute messages. In particular, we are interested in determining whether considering interactions between different nodes participating in a hyperedge is necessary. To this end, we consider an alternative scheme in which we first compute a hyperedge representation using aggregation of all adjacent nodes, using their qualifier information, *i.e.*

$$\mathbf{q}_e^{(t)} = \textsc{Agg}' \left( \left[ \mathbf{h}_e^{(t)}, \mathbf{r}_{\tau_1} \oplus \mathbf{h}_{n_1}^{(t)}, \mathbf{r}_{\tau_2} \oplus \mathbf{h}_{n_2}^{(t)}, ... \right] \right).$$

Table 2: Results on the Random Bugs dataset when applying HEAT on a graph dataset representing edges as (binary) edges.

|  | Joint Localisation & Repair | Localisation | Repair |
|---|---|---|---|
| GNN (Allamanis et al., 2021) (rerun) | 69.8 | 79.6 | 83.4 |
| GREAT (Allamanis et al., 2021) (rerun) | 65.6 | 74.4 | 81.8 |
| HEAT (on binarised hypergraphs) | 71.6 | 79.3 | 85.5 |
| HEAT (on hypergraphs) | 76.5 | 83.1 | 88.5 |

In our experiments, we use a Deep Set (Zaheer et al., 2017) model to implement $\textsc{Agg}'$. We then compute messages for each node $n_i$ using a single linear layer $W$, *i.e.*

$$\mathbf{m}_{e \to n_i}^{(t)} = ReLU\left(W \cdot [\mathbf{r}_{\tau_i} \oplus \mathbf{h}_{n_i}^{(t)}, \mathbf{q}_e^{(t)}]\right).$$

The results indicate that this model variant is still stronger than the GNN and GREAT architectures, but that HEAT profits from explicitly considering the relationships of nodes participating in a hyperedge.

Next, we analyse the importance of using qualifier information in our model. To this end, we consider a variant of HEAT in which we remove from Eq. 5 the qualifier information $\mathbf{r}_{\tau_i}$, *i.e.* to

$$\left[\mathbf{m}_e^{(t)}, \mathbf{m}_{e \to n_1}^{(t)}, \mathbf{m}_{e \to n_2}^{(t)}, ...\right] = \text{MHA}\left(\left[\mathbf{h}_e^{(t)}, \mathbf{h}_{n_1}^{(t)}, \mathbf{h}_{n_2}^{(t)}, ...\right]\right).$$

The results clearly indicate that the qualifier-as-position scheme used in HEAT is crucial for good performance. In particular, it indicates that the qualifier information contained in the data is very valuable, and emphasises the importance of considering *qualified* hypergraphs.

We also considered using a more expressive aggregation mechanism for messages, replacing the max pooling we use to implement $\textsc{Agg}$. Specifically, we use a multi-head attention between the current node state (as queries in the attention mechanism) and the messages computed for all adjacent hyperedges (appearing as keys and values). This is reminiscent of graph attention networks (Veličković et al., 2018), which use an attention mechanism to determine the respective importance of binary edges when aggregating messages. In our experiments, this performed slightly worse than the aggregation using a simple max, while being substantially more memory- and compute-intensive.

Next, we analyse whether the representational capacity is improved by the feedforward network (Eq. 4) in the node and edge update. Removing these substantially reduces the number of parameters of the model. The results indicate that these intermediate, per-representation computation steps add valuable capacity to the model. This is especially apparent on the PyPIBugs dataset.

Another ablation we consider is a model variant in which we do not use evolving edge states, but instead re-use $\mathbf{h}_e^{(0)}$ on all HEAT layers. We observe a similar or slightly reduced performance on the joint localisation and repair. This suggests that updating the edge state provides valuable representational capacity to the model. Edge states, can be seen as the `[CLS]` token in traditional transformers, providing "scratch space" for storing intermediate information for each hyperedge.

Finally, we assess the importance of the underlying data representation, investigating whether our representation of programs as hypergraphs (differing from the PyBugLab baseline) alone explains the performance improvement. For this, we convert the generated hyperedges to (binary) edges and use HEAT to learn from these (binary) graphs. This experiment intents to disentagle the effects of the different data representation from the effects of HEAT's architecture. The results, shown in Table 2, indicate that even on binarised hypergraphs, HEAT outperforms the baseline GNN and GREAT models (especially on accuracy of choosing program repairs), even though large parts of its architecture are not properly utilised by the data. However, HEAT also takes advantage of the more expressive and compact data representation.

## 4.2 HEAT for Knowledge Graph Completion

Knowledge graphs (KGs) can be accurately represented as typed and qualified hypergraphs. We now focus on link prediction over a hyper-relational KG, which can be viewed as completing a knowledge base by additional likely facts.

**Dataset** Following the discovery of test leaks and design flaws by Galkin et al. (2020) in common benchmark datasets such as WikiPeople (Guan et al., 2019) and JF17K (Wen et al., 2016) we chose one of the variations of the new WD50K dataset presented there – WD50K (100). It is derived from Wikidata, containing 31k statements, all of which use some qualifier. To model a qualified triple statement of the form $(s, r, o, Q)$ with $Q$ a set of qualifier/entity pairs $(qr_i, qv_i)$, we create a single hyperedge with special qualifiers `src` (resp.) `obj` for the source and object of the relation, *i.e.* $(r, \{(\texttt{src}, s), (\texttt{obj}, o)\} \cup Q)$. Using the example of Fig. 1 of Galkin et al. (2020), the fact that Einstein studied mathematics at ETH Zurich in his undergraduate is expressed as the hyperedge: `EducatedAt(src:`$n_{\texttt{Einstein}}$`, obj:`$n_{\texttt{ETH Zurich}}$`, degree:`$n_{\texttt{Bachelor}}$`, major:`$n_{\texttt{Mathematics}}$`)`.

In the released WD50K dataset, raw Wikidata identifiers (*e.g.* `P69`, `Q937`, etc.) are used to refer to entities and relation names. We enrich the dataset by additionally retrieving natural language information for these entities from Wikidata (*e.g.* replacing `P69` by "educated at") and allow the models to consume this information, to encourage similar treatment of similarly named relationships and entities.

**Model Architecture** We modified the open-source release of StarE by Galkin et al. (2020), replacing the GNN-based StarE encoder by HEAT. We used a single layer of HEAT with embedding size of 100 and a single layer of the Transformer used for calculating the final predictions[1]. Apart from some training parameters (see below), hyperparameters (dropout, *etc.*) remain as documented in Appendix C of Galkin et al. (2020). Since our goal is to evaluate the effectiveness of HEAT, the remainder of the model is identical to the original implementation of Galkin et al. (2020): queries for a relation $j$ are represented as the concatenation of the entity embeddings, relation embedding, and qualifier embeddings, as shown in Fig. 3 of Galkin et al. (2020). These are then passed through a Transformer block and a fully-connected layer to obtain the probability over each entity being a possible object of the relation.

To process the additional natural language information about relations and entities we extracted (see above), we build a vocabulary using byte pair encoding (BPE), and then embed the individual tokens and use sum pooling to obtain initial node and relation representations. We evaluated variations of both the original StarE model and our HEAT-based variant using this information, see below.

**Training and evaluation** Training is performed as in Galkin et al. (2020) using binary cross entropy with label smoothing. In this link prediction task, a matching score is calculated for all possible relation objects given source, relation and qualifier-entity pairs (Dettmers et al., 2018, p.3). We trained our model for 1k epochs with a learning rate of 0.0004 and batch size of 512. Hyperparameters were fine-tuned manually, using the provided validation set in the StarE implementation. For direct comparison with Table 3 of Galkin et al. (2020) we report mean reciprocal rank (MRR) and hits at 1 and 10 (H@1, H@10) matching the original evaluation setting. We train and evaluate all models using 5 random seeds and report standard deviations.

**Results** Table 3 shows the results of our evaluation on WD50K (100). We reran the original StarE implementation, both to validate our setup and to obtain standard deviations. First, we observe that using HEAT instead of the original GNN-based StarE encoder improves results, without any further changes to the architecture. We expect that recent orthogonal work improving StarE by Yu & Yang (2021) would similarly improve with our model.

Next, we consider the influence of using natural language information extracted from Wikidata. We note that the original StarE encoder does not profit from this information. In contrast, the HEAT-based model slightly improves results, even though most words in the extracted data are extremely sparse. Finally, we evaluate the less expressive message aggregation scheme of max pooling (as discussed in Sec. 4.1). Here, we see that its performance is only marginally worse.

---

[1]Figure 3 of Galkin et al. (2020), bottom rectangle.

Table 3: Link prediction results on WD50K (100) of Galkin et al. (2020). Standard deviations are obtained over 5 different seeds.

| Model | MRR | H@1 | H@10 |
|---|---|---|---|
| STARE (GNN)$^\dagger$ (Galkin et al., 2020) | $0.654_{\pm 0.002}$ | $0.586_{\pm 0.002}$ | $0.777_{\pm 0.002}$ |
| STARE (GNN) – with NL information | $0.653_{\pm 0.003}$ | $0.586_{\pm 0.003}$ | $0.774_{\pm 0.005}$ |
| STARE (HEAT) – AGG $\triangleq$ max, no NL information | $0.666_{\pm 0.003}$ | $0.605_{\pm 0.004}$ | $0.779_{\pm 0.002}$ |
| STARE (HEAT) – AGG $\triangleq$ CrossAtt, no NL information | $0.666_{\pm 0.003}$ | $0.599_{\pm 0.004}$ | $0.787_{\pm 0.003}$ |
| STARE (HEAT) – AGG $\triangleq$ CrossAtt, with NL information | $0.667_{\pm 0.003}$ | $0.601_{\pm 0.003}$ | $0.789_{\pm 0.001}$ |
| Hy-Transformer* (Yu & Yang, 2021) | 0.699 | 0.637 | 0.812 |

$^\dagger$ Rerun of Galkin et al. (2020) implementation with 5 seeds.
\* Current state-of-the-art.

## 5 Related Work

We review closely related techniques for learning on hypergraphs, and then briefly discuss some particularly relevant work from the application areas of learning on code and knowledge bases.

**Hypergraph learning** We broadly classify hypergraph learning into three approaches: hyperedge expansion, spectral methods, and spatial (message passing-based) methods. Compared to HEAT, none of the existing methods can directly work on typed and qualified hypergraphs.

In the first class, hypergraphs are transformed into (binary) graphs and then a standard GNN is applied on the resulting graph. This approach is taken by Agarwal et al. (2006), who use clique and star expansion (representing each hyperedge as a full or a star graph respectively), Yadati et al. (2019), who represent each hyperedge by a simple weighted edge whose endpoints can be further connected with weighted edges to mediator nodes and Yang et al. (2020), who create a node for each pair of incident nodes and hyperedges and connect those stemming from the same node or hyperedge. These approaches can re-use existing GNN architectures, but at the cost of a significantly increased number of edges.

The next class of methods aims at generalising the concepts of graph Laplacians and spectral convolutions to the domain of hypergraphs. Feng et al. (2019) use the observations that nodes in the same hyperedge should not differ much in embedding to define a normalised hypergraph Laplacian. A similar approach has been employed by Fu et al. (2019), who utilise the hypergraph $p$-Laplacian. Bai et al. (2021) extend Feng et al. (2019) where the node-edge incidence matrix is weighted via an attention mechanism. Such methods are usually applied to transductive learning tasks and either do not fully support typed or qualified hyperedges or are limited to a fixed number of hyperedge types or nodes in a hyperedge. This makes them inapplicable in our setting.

The final class of methods generalises the concept of neural message passing to hypergraphs. HyperSAGNN (Zhang et al., 2020) is a self-attention based neural network, capable of handling variable-sized hyperedges, but has been developed for the purposes of hyperedge prediction and is not directly applicable for node classification. HyperSAGE (Arya et al., 2020) performs convolution in two steps: nodes to hyperedges and hyperedges to nodes, where the aggregation during the node and hyperedge message passing step is a power-mean function, but it suffers from poor parallelisation and other practical issues (Huang & Yang, 2021, p. 2). Chien et al. (2021) propose AllSet, a message passing scheme based on representing hyperedges as sets and aggregations using permutation-invariant functions. AllSet can provably subsume a substantial number of previous hypergraph convolution methods. Chien et al. (2021) propose two implementations of AllSet, using DeepSets (Zaheer et al., 2017) and Transformers (Vaswani et al., 2017). This work is most similar to ours, but does not consider the setting of qualified hyperedges. A natural consequence is that the model then computes a single "message" per hyperedge, whereas HEAT computes different messages for each participating node (which only is required when nodes play different roles in the hyperedge). In Sec. 4.1, we consider two relevant ablations. These show that the use of qualifier information is crucial for good results in the PyBugLab setting, and that our message computation based on multihead attention is stronger than a DeepSet-based alternative.

To the best of our knowledge, we are the first to focus on processing typed and qualified hyperedges: two hyperedges with the same element set can be different if their type does not match and elements in a hyperedge can have different qualifiers (roles). Furthermore, our architecture is not restricted to a fixed number of hyperedge types/qualifier and can generalise to types/qualifiers unseen during train time.

**Knowledge Graph Completion**   Knowledge graph (KG) completion has emerged due to the incompleteness of KGs (Ji et al., 2021). Embedding-based models (Bordes et al., 2013; Schlichtkrull et al., 2018; Shi & Weninger, 2017) first learn a low-dimensional embedding and then use it to calculate scores based on these embeddings and rank the top $k$ candidates. HEAT is a variation of an embedding-based model, but we focus on hyper-relational KGs. Other, non-embedding-based approaches also exist, *e.g.* reinforcement learning (Xiong et al., 2017) or rule-based ones (Rocktäschel & Riedel, 2017). Since we do not use such techniques, we omit discussing them here and refer the reader to Ji et al. (2021, §IV.A).

Similarly to hypergraph learning, other works model hyper-relational KGs by simplifying qualified relations to simpler representations: Wen et al. (2016) use clique expansion, which can be costly, (Fatemi et al., 2020) represent hyper-relational facts as $n$-ary relations, but do not have explicit source/object of the relation and instead of considering the qualifier of an entity as in HEAT, their model only considers the position of an entity within the relation; Guan et al. (2019) break $n$-ary facts into $n + 2$ qualifier/entity pairs[2], making qualifier/entity pairs indistinguishable to "standard" $(s, rel, o)$ triples. These expansion-based approaches cannot leverage semantic information such as the interaction of different qualifier/entity pairs (Yu & Yang, 2021).

Closest to our work is STARE (Galkin et al., 2020). STARE uses a GNN-like convolution, consisting of several steps (cf. equations (5)-(7) of Galkin et al., 2020) on hyper-relational graphs to calculate updated entity embeddings, which are then fed through a Transformer module. In Sec. 4 we show that HEAT outperforms their GNN-based encoder.

**Learning on Code**   Over the last decade, machine learning has been applied to code on a variety of tasks (Allamanis et al., 2018a). A central theme in this research area is the representation of source code. Traditionally, token-level representations have been used (Hindle et al., 2012), but Allamanis et al. (2018b) showed that graph-level approaches can leverage additional semantic information for substantial improvements. Subsequently, Fernandes et al. (2019); Hellendoorn et al. (2020) showed that combining token-level and graph-level approaches yields best results. By using a relational transformer model over the code tokens Hellendoorn et al. (2020) overcomes the inability of GNN models to handle long-range interactions well, while allowing to make use of additional semantic information expressed as graph edges over the tokens. However, token-based representations do not provide unambiguous locations for annotating semantic information (*i.e.* edges) for non-token units such as expressions (*e.g.* `a+b` or `a+b+c`). Additionally, all these approaches have been limited to standard (binary) edges, making the resulting graphs large and/or imprecise (see Sec. 3). Our experiments with HEAT show that a suitable representation as typed and qualified hypergraph further improves over the combination of token-level and binary graph-level approaches.

## 6   Conclusions & Discussion

We introduced HEAT, a neural network architecture that operates on typed and qualified hypergraphs. To model such hypergraphs, HEAT combines the idea of message passing with the representational power of Transformers. Furthermore, we showed how to convert program source code into such hypergraphs. Our experiments show that HEAT is able to learn well from these highly structured hypergraphs, outperforming strong recent baselines on their own datasets. A core insight in our work is to apply the power of Transformers to several, overlapping sets of relations at the same time. This allows to concurrently model sequences and graph-structured data. We believe that this opens up exciting opportunities for future work in the joint handling of natural language and knowledge bases.

---

[2]+2 for the source and object qualifiers

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

# A    Packing Hyperedges for HEAT

Hyperedges in HEAT have a variable size, *i.e.* a variable number of nodes that participate in each hyperedge. For each hyperedge HEAT uses a multihead attention to compute the outgoing messages $\mathbf{m}_{e_i \rightarrow n_j}$. However, multihead attention has a quadratic computational and memory cost with respect to the size of the hyperedge.

A naïve solution would require us to pad all hyperedges up to a maximum size. However, this would be wasteful. On the other hand, we need to batch the computation across multiple hyperedges for efficient GPU utilisation. To achieve a good trade-off between large batch sizes (a lot of elements) and minimal padding, we consider a small set "microbatches". To pack the hyperedges optimally, we would solve the following integer linear program (ILP)

$$\min \sum_{j}^{1..k} \left( L_j^2 \cdot y_j \right) - \sum_{i} l_i^2$$

$$s.t.$$

$$y_j \in \{0,1\}, x_{ij} \in \{0,1\}, \forall i, j$$

$$\sum_{j}^{1..k} x_{ij} = 1, \forall i$$

$$\sum_{i}^{1..n} x_{ij} s_i \leq y_j L_j, \forall j,$$

where $y_j$ is a binary variable indicating whether "bucket" $j$ of size $L_j$ (*i.e.* belonging in the microbatch of size $L_j$) is used (*i.e.* contains any elements). Then, $l_j$ is the width of the $j$th hyperedge. $x_{ij}$ are binary variables for $i = 1..n$ and $j = 1..k$ indicating if the element $i$ is in bucket $j$ and $s_i$ the width of bucket $i$. The objective creates as few buckets as possible to minimise the wasted (quadratic) space/time needed for multihead attention over variable-sized sequences. The first constraint requires that each element is assigned to exactly one bucket and the second constraint that each used bucket is filled up to its capacity.

However, the complexity of this ILP is prohibitive. Instead, we resort in using a greedy algorithm to select the "microbatches" used. This is detailed in Algorithm 1.

---

**Algorithm 1** Greedy Hyperedge Packing into Microbatches

---

buckets ← [ ]
**for** h in sortedDescendingByWidth(hyperedges) **do**
    wasAdded ← False
    **for** bucket in buckets **do**
        **if** bucket.remainingSize ≥ h.width **then**
            bucket.add(h)
            wasAdded ← True
            break
    **if** not wasAdded **then**
        bucketSize ← smallestFittingMicrobatchWidth(h.width)
        newBucket ← createBucket(bucketSize)
        newBucket.add(e)
        buckets.append(newBucket)
**return** GroupBucketsToMicrobatches(buckets)

---

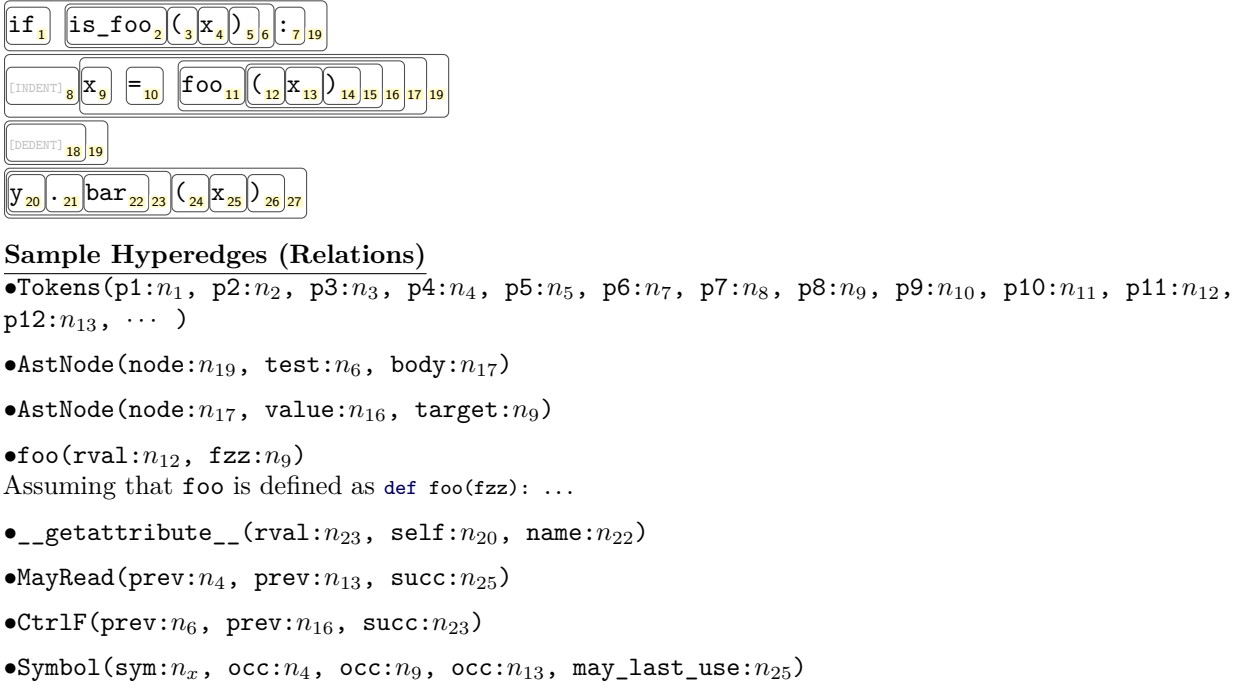

**Sample Hyperedges (Relations)**

•Tokens(p1:$n_1$, p2:$n_2$, p3:$n_3$, p4:$n_4$, p5:$n_5$, p6:$n_7$, p7:$n_8$, p8:$n_9$, p9:$n_{10}$, p10:$n_{11}$, p11:$n_{12}$, p12:$n_{13}$, $\cdots$ )

•AstNode(node:$n_{19}$, test:$n_6$, body:$n_{17}$)

•AstNode(node:$n_{17}$, value:$n_{16}$, target:$n_9$)

•foo(rval:$n_{12}$, fzz:$n_9$)
Assuming that `foo` is defined as `def foo(fzz): ...`

•__getattribute__(rval:$n_{23}$, self:$n_{20}$, name:$n_{22}$)

•MayRead(prev:$n_4$, prev:$n_{13}$, succ:$n_{25}$)

•CtrlF(prev:$n_6$, prev:$n_{16}$, succ:$n_{23}$)

•Symbol(sym:$n_x$, occ:$n_4$, occ:$n_9$, occ:$n_{13}$, may_last_use:$n_{25}$)

Figure 3: Sample relations for the synthetic snippet shown on the top. The AST nodes and tokens are wrapped in boxes and numbered appropriately in a preorder fashion. Only a few samples of the relations (mapped to hyperedges) are shown below.

## B  Code as Hypergraph Example

Fig. 3 shows a synthetic code snippet with all the token and AST nodes enclosed in boxes. Some sample relations are also shown. Finally, Fig. 4 shows a full hypergraph for the following code snippet

```
def foo(a, b):
  if a in b:
    a += 1
  return a * 2
```

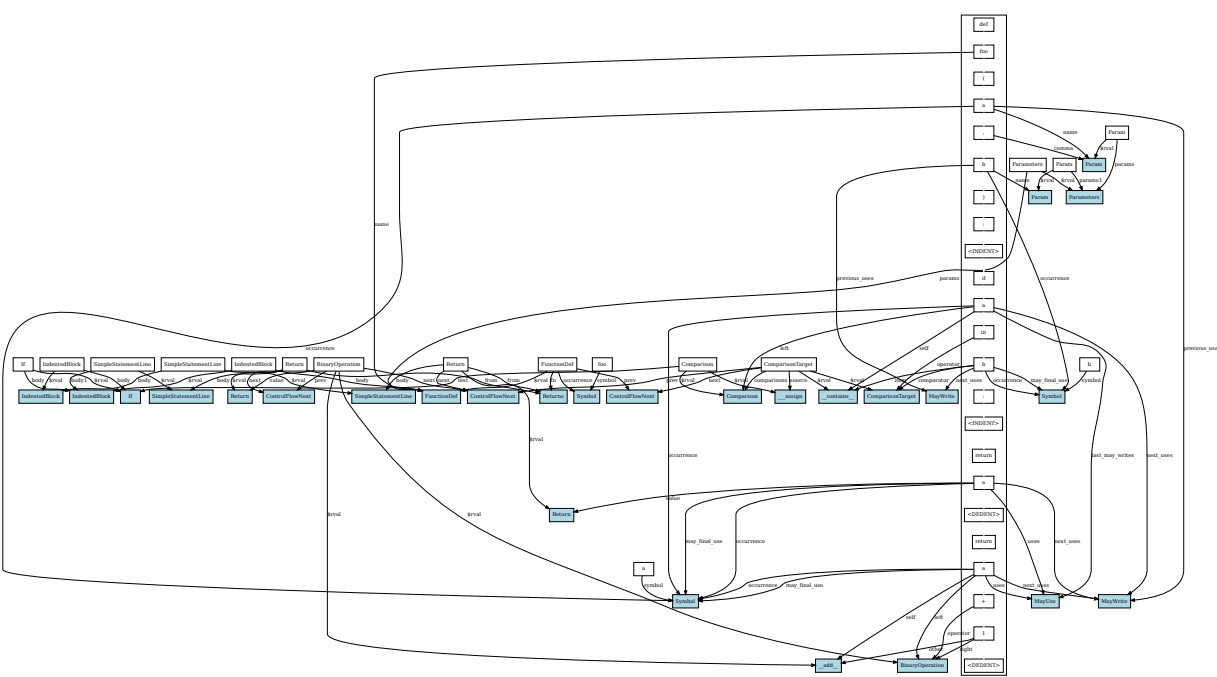

Figure 4: A full hypergraph sample for the snippet disc Hyperedges are denoted as shaded (blue) boxes. Best viewed on screen. The Tokens(·) hyperedge is omitted for clarity but the sequence of tokens is placed in the box (right).

