# OpenReview forum: "HEAT: Hyperedge Attention Networks"
_TMLR — Accepted by TMLR_

### Review · Reviewer_egRw · 2022-06-28

**Summary Of Contributions:**

The paper presents a neural architecture HEAT for encoding graphs with *typed* and *qualified* *hyperedges*.

This new architecture generalizes transformers by having the ability to represent relations rather than all-to-all interactions, generalizes standard message-passing GNNs by having the ability to represent *hyperedges* (rather than pairwise, binary, edges), and generalizes previous hyperedge-GNNs by having the ability to represent edges that are both *typed* and *qualified* (where the order of argument matters).

Such kinds of typed+qualified n-arry relations are very general and exist in many natural domains; in this paper, the authors demonstrate the importance of such relations in knowledge graphs and in programs, where the new approach provides significant gains over previous work.

**Broader Impact Concerns:**

no concerns

**Requested Changes:**

The difference between HEAT and baselines such as GREAT or GNN (Table 1) comes from two main sources:
1. The different representation of the data - pairwise relations (GNN, GREAT) vs. n-arry relations (HEAT).
2. The different neural architecture - that is, Transformer-style attention function, Multi-heads, LayerNorm, FeedForward layers, possibly more parameters, etc.

I wonder if the authors can perform some experiments to disentangle these two sources? For example, perform the experiments of Table 1 with the pairwise "topology" of the data (as GNNs), but with the HEAT "architecture" applied to pairwise relations?

In the other direction - how would the n-arry hyperedge "topology" (as in HEAT) would be processed by standard strong GNNs such as GIN/GAT/GATv2 ?


**Strengths And Weaknesses:**

## Strengths
+ The proposed architecture is very general and potentially applicable to a wide variety of domains
+ The finding that no existing architecture satisfies all the requirements of expressing typed+qualified+n-arry relations is important.
+ The architecture is based on components of the Transformer architecture but organized in a different way, which keeps it simple.
+ The significance of the results is convincing

## Weaknesses
- Additional tasks will further improve the paper. Even though typed+qualified+n-arry relations are very common in "the wild", I understand that it's difficult to find appropriate datasets that were processed, annotated / curated this way. Maybe the VarMisuse dataset (Allamanis et al., 2018) can be converted to n-arry relations? Can HEAT be applied to existing *pairwise* datasets such as VarMisuse? This is just a recommendation for further improving the paper, and not a mandatory change.

## Additional Comments and Questions
* The paper can be easily strengthened by comparing with additional types of out-of-the-box (for example, from PyTorch Geometric) GNN convolutions in Table 1 and Table 2.
* Minor: the authors claim that their architecture generalizes Transformers and message-passing GNNs. I think that by expressing arbitrary n-arry relations, HEAT also kind of generalizes arbitrary-shaped Conditional Random Fields (CRFs). What do the authors think about that? Is that a too strong claim?

## Summary
I am expecting to hear the authors' response to the section "Requested Changes" to secure my recommendation, but overall this is a good paper,
the made claims are supported by accurate, convincing, and clear evidence,
the paper is relevant to a very broad audience,
and I recommend acceptance.

---

> ### Comment · Reviewer_egRw · 2022-06-30
> **Additional minor comments**
>
> Additional minor presentation comments:
>
> * In Equation (1) and the (unnumbered) equation a few lines above it on page 2, both variables are called $\mathbf{m}^{(t)}$ (with a different subscript, which is a little confusing, and difficult to understand whether $MHA$ in Equation (2) is an implementation of the general $f_m$ function of Equation (0), or the $AGG$ function of Equation (1). $MHA$ "behaves" like $f_m$ (Equation (0)) but has the same notation as $AGG$ (Equation (1).
>
> * In the "Variations and Ablations" part on page 7, it is very difficult to match the description of the ablation to the appropriate line in Table 1.
>
> * I am uncomfortable with the ablations in Table 1 and Table 2 being performed on the test set rather than the validation set, since this is a kind of architecture/hyperparameter tuning on the test set.
>
> * Concrete examples will go a long way in improving the paper. In particular, examples which show the importance of each contribution separately (hyperedges, typed edges, qualified). Can the authors show (even cherry picked) examples where each of these components was important and helped their model predict accurately, while a baseline/ablation that does not have this component did not predict correctly?

---

> > ### Author Response · Authors · 2022-07-16
> > **Review Response 2/2**
> >
> > > In Equation (1) and the (unnumbered) equation a few lines above it on page 2, both variables are called  (with a different subscript, which is a little confusing, and difficult to understand whether  in Equation (2) is an implementation of the general  function of Equation (0), or the  function of Equation (1).  "behaves" like  (Equation (0)) but has the same notation as  (Equation (1).
> >
> > There seems to be a misunderstanding here. Eq. (1) (and the unnamed one you refer to as Eq. (0)) define the aggregation of messages in GNNs, whereas Eq. (2) is part of the definition of Transformers. While there are some similarities, these do not behave the same (in particular; Transformers do not take the edge topology into account; and (most) GNNs compute messages indepedently of each other). We will think about how to update the notation to avoid confusion.
> >
> > > Concrete examples will go a long way in improving the paper. In particular, examples which show the importance of each contribution separately (hyperedges, typed edges, qualified). Can the authors show (even cherry picked) examples where each of these components was important and helped their model predict accurately, while a baseline/ablation that does not have this component did not predict correctly?
> >
> > We have decided against including such cherry-picked examples as the reason for models making a single right (or wrong) prediction is hard to pin down. It would be preferable to have a finer-grained split of the test dataset to illustrate these issues in detail.
> > The one such observation that we can support quantitatively is that our HEAT-based bug detector model substantially outperforms baselines on the "argument swapping" class of bugs. Concretely, both the GNN-based and GREAT-based model only reach an accuracy of ~20% on these kinds of bugs, whereas our HEAT-based model reaches an accuracy of ~80%.
> > In this case, the ability of modelling an argument list, together with the names of relevant parameters, as a single hyperedge is clearly beneficial, as all of these components can directly interact in a single multihead attention step.

---

> ### Author Response · Authors · 2022-07-16
> **Review Response 1/2**
>
>
> Thank you for your kind and detailed review. We will integrate your detailed minor comments in the next revision of our paper, and only reply to some points here.
>
> > The difference between HEAT and baselines such as GREAT or GNN (Table 1) comes from two main sources:
> > - The different representation of the data - pairwise relations (GNN, GREAT) vs. n-arry relations (HEAT).
> > - The different neural architecture - that is, Transformer-style attention function, Multi-heads, LayerNorm, FeedForward layers, possibly more parameters, etc.
> >
> > I wonder if the authors can perform some experiments to disentangle these two sources? For example, perform the experiments of Table 1 with the pairwise "topology" of the data (as GNNs), but with the HEAT "architecture" applied to pairwise relations?
> >
> > In the other direction - how would the n-arry hyperedge "topology" (as in HEAT) would be processed by standard strong GNNs such as GIN/GAT/GATv2 ?
>
> We are now running an experiment to answer the first question (how would HEAT perform on binary graphs), and will report the results as soon as training and evaluation have finished.
> We can note that we have investigated a wide range of GNN variants, introducing features from the Transformer architecture (such as residual block structure, feed-forward layers, layer normalization, _etc._). While we have not prepared these results for publication, we can anecdotally report that these changes do not suffice to boost performance substantially.
>
> Regarding the "other direction", we are not entirely sure what you mean: there are many, quite different ways of representing hyperedges as binary edges (see Sect. 5 of our submission), and so there are many potential choices here that could be explored. Earlier work (e.g., Feng et al. (2019) [a] and Chien et al. (2021) [b]) have provided empirical evidence that direct modelling of hyergraphs outperforms a pipeline that first expands hyperedges and then applies a standard in GNN. In particular, Chien et al. (2021) [b] discuss how a number of different hypergraph-specific models can be understood as performing implicit hyperedge expansion, and then shows that these models are a special case of their AllSet model. As we discuss in our related work section, the AllSet model can be viewed as a simplification of HEAT (as AllSet does not consider qualified hyperedges).
>
> We also note that as a baseline we use the GNN architecture of Allamanis _et al._ (2021) which has shown strong performance. Brockschmidt (2020) [c] shows that the MLP-based GNN performs similarly or better compared to strong baselines such as GAT or GIN.
>
> > Minor: the authors claim that their architecture generalizes Transformers and message-passing GNNs. I think that by expressing arbitrary n-arry relations, HEAT also kind of generalizes arbitrary-shaped Conditional Random Fields (CRFs). What do the authors think about that? Is that a too strong claim?
>
> There is indeed some resemblance of factor graphs to hypergraphs (factors are analogous to hyperedges and variables are analogous to nodes). However, we think that it would be a too strong claim that HEAT generalizes CRFs:
> Although the self-attention mechanism (Fig. 2(a)) can be used to represent the factors in a CRF, it is unclear how the rest of the network would map to the standard definition of a (conditional) factor graph, since the energy function takes a much more complicated form.
>
> References
> ---
> [a] Yifan Feng, Haoxuan You, Zizhao Zhang, Rongrong Ji, and Yue Gao. "Hypergraph neural networks." AAAI Conference on Artificial Intelligence 2019.
>
> [b] Eli Chien, Chao Pan, Jianhao Peng, and Olgica Milenkovic. "You are AllSet: A multiset function framework for hypergraph neural networks."" arXiv preprint arXiv:2106.13264.
>
> [c] Marc Brockschmidt. "GNN-FiLM: Graph neural networks with feature-wise linear modulation." International Conference on Machine Learning 2020.

---

> > ### Comment · Reviewer_egRw · 2022-07-22
> > **Any news?**
> >
> > Dear authors,
> > I agree that my "other direction" suggestion may not make sense, or this was not a concrete question.
> >
> > Is there any news regarding the question of "how would HEAT perform on binary graphs"?
> > Is there an estimated time for the experiment to complete?
> > I would like to get a better understanding of the contribution of each component.

---

> > > ### Author Response · Authors · 2022-07-22
> > > **Re: Any news?**
> > >
> > > Our training run is currently at epoch 26 (out of 100 maximum) but it has started to overfit, so it may not need all 100 epochs (we are using early stopping with patience of 10 epochs). Note that with the 'binarisation', the number of hyperedges in the training data is roughly ~400M and that we currently do not have access to our original, more powerful GPU resources. A single training epoch in this setting takes ~10 hours.
> > >
> > > From the training/validation metrics we can observe, we note that the accuracy of fixing "argument swapping" bugs is substantially lower than for the HEAT model on the full hypergraphs: it dropped from ~80% to ~67%, even on the training data, illustrating that it is harder for the model to fit the data. On the other bug classes, results are slightly worse than on the "full" model.

---

> > > > ### Author Response · Authors · 2022-08-16
> > > > **Thank you for the patience**
> > > >
> > > > Dear Reviewer,
> > > >
> > > > Thank you for the patience and understanding of our situation. The training and evaluation is finally over. Results are in the table below (Random Bugs dataset). Original HEAT model results repeated for comparison.
> > > >
> > > > | Model |Joint|Loc.|Repair|
> > > > |---------|-------|------|---------|
> > > > |HEAT  |  76.5  | 83.1 |  88.5  |
> > > > |HEAT-binary-edges |  71.6  | 79.3 |  85.5  |
> > > >
> > > > Clearly, having hyperedges instead of binary edges (that describe an equivalent sample) is beneficial both for localisation and repair.

---

> > > > > ### Comment · Reviewer_egRw · 2022-08-16
> > > > > **Thank you**
> > > > >
> > > > > Thank you for the additional experiments.
> > > > > The additional experiments clearly show the benefit of the architecture and the benefit of the typed+qualified n-arry edges.
> > > > >
> > > > > I recommended acceptance before, and I keep my recommendation.

---

### Review · Reviewer_KWMM · 2022-07-06

**Summary Of Contributions:**

In this work the authors introduce a novel model for hypergraph representation called HEAT, which combines ideas from message passing neural networks (MPNN) and transformers. HEAT learns embeddings for the hyperedge, each node, and the qualifiers of the hyperedge which are updated by several rounds of message-passing. The connection to transformers is that the message are calculated via multi-headed attention, where the qualifier representation takes the place of the positional encodings in a standard transformer. Messages from all edges that a node participates in are then aggregated using a permutation-invariant function, at which point the standard transformer update is applied (a linear combination of current node embedding and the message is passed through a layer normalization, combined with a feed-forward network projection of itself and layer-normalized again). The authors evaluate this architecture on two tasks: knowledge base completion and bug detection and repair, finding that it outperforms reasonable baselines in both tasks.

**Broader Impact Concerns:**

None.

**Requested Changes:**

I do not have any significant changes to request.

Typos:
1: "... profit from modelling as ..." -> "... profit from being modeled as ..."
2: Consider assigning the edge to $e$, eg. $e = (n_i, \tau_i, n_j)$, and make the subsequent $\tau$ into $\tau_i$.
6: "... requires to combine ambiguous information with reasoning capabilities towards detecting bugs ..." -> "... requires combining ambiguous information with reasoning capabilities capable of detecting bugs ..."
8: "... replacing the the GNN-based ..." -> "... replacing the GNN-based ..."
8: Could not parse "... and embedding of qualifiers, entity pairs of the relation $j$."
8: "... and a fully-connected layers to ..." -> "... and a fully-connected layer to ..."

**Strengths And Weaknesses:**

Strengths:
+ Very well written, exposition and mathematical details were clear
+ Strong scientific rigor with experimentation (eg. rerunning baselines when there was unavoidable dataset discrepancies, as well as providing them similar benefits as HEAT where possible)
+ Thorough ablation study evaluating architectural design choices (eg. aggregation function, splitting tokens, use of qualifier embeddings)
+ The model is well-motivated, from both a mathematical and practical standpoint
+ Computational implementation details are included, to the extent that reproducing the model should be possible
+ The evidence presented does support the author's claims regarding the benefits of the proposed model

Weaknesses:
- The model is only evaluated on two datasets, however:
  (1) Evaluating on another bug-detection dataset may require significant effort, due to programming language, and (while I am not overly familiar with bug-detection datasets) PyBugLab seems to be the best choice for such an evaluation.
  (2) There are admittedly few hyperrelational datasets available, and WD50K is the recommended in Galkin et al. as other choices have various issues.
While these limitations are understandable, it also means that it is difficult to draw broader conclusions. For example, no single pooling method performs best in all scenarios, so a practitioner employing it would have to do some minor architecture search.

- A more thorough comparison to baselines would strengthen the claims, and make the paper more complete.
  (1) For bug detection, the authors mention that a comparison to recent large language models is left for future work. It is likely that adapting such models to this task would take a reasonable amount of effort, however such an evaluation on it's own may not be significant enough to stand on its own as a publication, and thus the concern is that such a comparison is never made and further effort may be expended on tasks which are unwittingly already solved by other approaches.
  (2) For KB Completion, the authors mention the recent work of Yu & Yang (2021), and claim that the advantages provided in that work are mostly orthogonal to those provided by HEAT, suggesting that the two might work synergistically. It must be noted, however that the Hy-Transformer architecture of Yu & Yang greatly outperforms HEAT on WD50K (100), with MRR 0.699, H@1 0.637, and H@10 0.812. For completeness it would seem reasonable to include this baseline in the table, and for completeness actually demonstrate the suggested synergistic benefits of HEAT and Hy-Transformer (which, it seems, may also lead to additional benefits on the bug-detection task, now that it is framed as hypergraph completion).

Questions:
- Second paragraph, page 8, it is stated that the additional computational depth provided by the feedforward network improves *generalization*. One would assume that adding this significant number of parameters would have more of an effect on overall model capacity, perhaps at the expense of (relative) generalization performance - is it really generalization that is improved, or overall representational capacity?
- How much does using sub-tokens (as described in the middle of page 4) help with the bug detection task?

---

> ### Author Response · Authors · 2022-07-16
> **Review Response**
>
>
> Thank you for your review and engaging with our submission. We will update our submission to reflect the typos and wording problems you have identified.
>
> > For bug detection, the authors mention that a comparison to recent large language models is left for future work.
>
> We have avoided this as we lack access to such models (and computational resources to train them from scratch). In particular, as we would be interested in a non-generative task, we would require access to intermediate activations of a model, requiring deeper access than just an API giving completions.
>
> >  For completeness it would seem reasonable to include this baseline in the table, and for completeness actually demonstrate the suggested synergistic benefits of HEAT and Hy-Transformer (which, it seems, may also lead to additional benefits on the bug-detection task, now that it is framed as hypergraph completion).
>
> As far as we know, the authors of Hy-Transformer have not released their implementation. While we believe we could re-implement the model changes based on the published paper (though of course there's a risk that crucial details required for the impressive results are not documented), we expect that this would require substantial extra work, especially to determine good hyperparameters for the joint architecture. Consequently, we would instead propose to just include the Hy-Transformer numbers in our result table, to clarify that these are the current state of the art.
>
> We would also note that we have not framed bug detection as hypergraph completion, but simply use HEAT to produce representations that are then trained together with the subnetworks described in Allamanis et al. (2021).
>
> > Second paragraph, page 8, it is stated that the additional computational depth provided by the feedforward network improves generalization. One would assume that adding this significant number of parameters would have more of an effect on overall model capacity, perhaps at the expense of (relative) generalization performance - is it really generalization that is improved, or overall representational capacity?
>
> We drew this conclusion based on the much more pronounced drop in localization performance of the "HEAT - without FFN" ablation on the PyPIBugs dataset compared to the results on Random Bugs. This difference points towards generalization (as the PyPIBugs dataset is slightly differently distributed), but indeed, our experiments are insufficient to robustly support the claim. We will rephrase this sentence to simply state that representational capacity is improved.
>
> > How much does using sub-tokens (as described in the middle of page 4) help with the bug detection task?
>
> We do not have any data to definitively answer this question and would like to avoid running a costly experiment. However, given past work, subtokenization should not hurt performance.
>
> Specifically, subtokenization is a common technique used in machine learning for code, due to the fact that variable/function names are rare, but a combination of common subtokens. Othwerwise a large percent of identifiers would be represented as `UNK` (e.g., the fact that `read_foobar` and `read_wuzzlebuzzle` both contain `read` is a strong indicator about their functionality, nonwithstanding the lack of knowledge about `wuzzlebuzzle`). Given, this, the use of subtokens is common is such works and is important for HEAT for representing an open vocabulary of qualifiers (_e.g.,_ function parameters).

---

### Review · Reviewer_MmCu · 2022-07-07

**Summary Of Contributions:**

This paper proposes to use self-attentional neural networks to learn hypergraph-based representations. Different from traditional approaches that mainly consider binary relations, the proposed approach, HEAT, models more complicated hyperedges and the associated qualifier (role) information of the edges, which are proved to be crucial in empirical ablations. HEAT adopts the transformer arch to compute node and edge representations while modifying the multi-head attention based on the graph structure. On code bug detection/repair and knowledge graph completion experiments, HEAT outperforms strong baselines that are based on GNNs.


**Broader Impact Concerns:**

This paper presents a general model for hypergraph modelling and applies it to bug detection/repair and KG completion, I don’t have concerns on its ethical implications.


**Requested Changes:**

*An important detail to secure my recommendation for acceptance:*

When replacing layers from the baseline with the HEAT layer, do you strictly control the model size to be the same for a fair comparison? I didn’t find this information in the paper but I feel it is necessary to clarify it.


**Strengths And Weaknesses:**

*Strengths:*

1. While the use of transformers for hypergraph modelling is not new (Chien et al. 2021), incorporating qualifier information like positional embeddings is a nice contribution and ablations in Table 1 show that it is very effective.
2. Empirical results are strong.

*Weaknesses:*

I do not identify significant weaknesses.

---

> ### Author Response · Authors · 2022-07-16
> **Review Response**
>
> Thank you for your kind review.
>
> > When replacing layers from the baseline with the HEAT layer, do you strictly control the model size to be the same for a fair comparison? I didn’t find this information in the paper but I feel it is necessary to clarify it.
>
> Both HEAT and GREAT use the same transformer layers (shapes, number of heads, etc) for a fair comparison. However, HEAT has the additional capacity needed for the qualifier embeddings.

---

### Decision · Action_Editors · 2022-08-17

**Recommendation:** Accept as is

**Comment:**

The reviewers liked the paper and were satisfied with the authors' responses to their questions. Given this, I am delighted to recommend acceptance. Please make sure to incorporate the feedback in the reviews in the final version.